

# Manipulating network connectance by altering plant attractiveness

Laura Russo[1] and Jane C. Stout[2]

[1] Department of Ecology and Evolutionary Biology, University of Tennessee, Knoxville, TN, USA
[2] Department of Botany, University of Dublin, Trinity College, Dublin, Ireland

## ABSTRACT

**Background:** Mutualistic interactions between plants and their pollinating insects are critical to the maintenance of biodiversity. However, we have yet to demonstrate that we are able to manage the structural properties of these networks for the purposes of pollinator conservation and preserving functional outcomes, such as pollination services. Our objective was to explore the extent of our ability to experimentally increase, decrease, and maintain connectance, a structural attribute that reflects patterns of insect visitation and foraging preferences. Patterns of connectance relate to the stability and function of ecological networks.

**Methods:** We implemented a 2-year field experiment across eight sites in urban Dublin, Ireland, applying four agrochemical treatments to fixed communities of seven flowering plant species in a randomized block design. We spent ~117 h collecting 1,908 flower-visiting insects of 92 species or morphospecies with standardized sampling methods across the 2 years. We hypothesized that the fertilizer treatment would increase, herbicide decrease, and a combination of both maintain the connectance of the network, relative to a control treatment of just water.

**Results:** Our results showed that we were able to successfully increase network connectance with a fertilizer treatment, and maintain network connectance with a combination of fertilizer and herbicide. However, we were not successful in decreasing network connectance with the herbicide treatment. The increase in connectance in the fertilized treatment was due to an increased species richness of visiting insects, rather than changes to their abundance. We also demonstrated that this change was due to an increase in the realized proportion of insect visitor species rather than increased visitation by common, generalist species of floral visitors. Overall, this work suggests that connectance is an attribute of network structure that can be manipulated, with implications for management goals or conservation efforts in these mutualistic communities.

# INTRODUCTION

Networks of mutualistic interactions are the architecture of biodiversity (*Bascompte & Jordano, 2007*). These mutualisms provide key ecosystem services, such as pollination, and produce predictable patterns of interaction structure (*Schleuning, Fründ & García, 2015*; *de Santiago-Hernández et al., 2019*; *Arceo-Gómez et al., 2020*). Significant exploratory

Corresponding author
Laura Russo, lrusso@utk.edu

experimental work has shown that interactions between flowering plants and their insect visitors result in consistent, reproducible patterns over space and time (*Lopezaraiza-Mikel et al., 2007*; *Larue, Raguso & Junker, 2016*; *Brosi, Niezgoda & Briggs, 2017*; *Russo et al., 2019*; *Maia, Vaughan & Memmott, 2019*; *Biella et al., 2020*; *Bain et al., 2022*).

Experimental manipulation and control of network structures is important to avoid non-target effects of management (*Cagua, Wootton & Stouffer, 2019*). Whereas manipulating a single species at a time can sometimes result in the loss of beneficial biodiversity and other undesirable outcomes (*Prabhaker et al., 2011*; *Weathered & Hammill, 2019*; *Gagic et al., 2019*), network control theory is a strategy to determine whether a system can be managed from an initial stable community composition toward a desired final state in finite time, given management inputs (*Liu, Slotine & Barabasi, 2011*; *Cagua, Wootton & Stouffer, 2019*). Developing management strategies with a network framework allows for multispecies management and improved outcomes (*Garrison et al., 2012*).

However, the extent to which we can deliberately manipulate network attributes of ecological systems is less well studied (*Dormann & Blüthgen, 2017*; *Valdovinos, 2019*). For example, previous work has shown that network connectance can be experimentally increased (*Russo & Shea, 2016*), but the degree to which this is repeatable and whether it can also be decreased, is unknown. More broadly, it has been proposed that these network structures may relate to stability, function, and management objectives (*Tylianakis et al., 2010*; *Cagua, Wootton & Stouffer, 2019*). Connectance is thought to be an attribute of network structure that directly relates to stability and function (*Tylianakis et al., 2010*), and reflects patterns of insect visitation and foraging preferences in a plant-flower visitor network (*Blüthgen et al., 2008*). On the other hand, the role of connectance in network function is controversial. While potentially useful for directing management objectives due to the ease in which it is measured and its relationship to interaction redundancy (*Tylianakis et al., 2010*), it is not clear that connectance has a positive effect on conservation value (*Heleno, Devoto & Pocock, 2012*). Despite this controversy, the capacity of managers to directly control network attributes has not been clearly demonstrated and it is important to establish that these networks are empirically controllable before recommending that any given attribute could be critical to the conservation of mutualisms and biodiversity.

Because bipartite plant-flower visitor networks have been so thoroughly empirically and theoretically studied, they provide an excellent opportunity to implement network control concepts (*Forbes & Northfield, 2017*; *Cagua, Wootton & Stouffer, 2019*). As an empirical test of our degree of control, our objective was to determine whether we would be able to manage a plant-flower visitor network to increase, decrease, or maintain network connectance. Although network connectance is well-studied, the drivers of patterns of connectance in networks are less well understood. Connectance is an ideal target for initial experimental manipulations of network structure because it is directly tied to the number of interactions formed by each species in the network. If the number of interactions can be manipulated by increasing or decreasing attractiveness, so can connectance. In a
flower-visitor network, the number of interactions is directly related to the attractiveness and resource quality of the plant species.

We designed and implemented a 2 year field experiment to generate 31 plant-flower visitor networks in four treatment categories. Our goal was to illustrate the degree to which we could experimentally manipulate network connectance by increasing (through chemical fertilizer) and decreasing (through chemical herbicide) connectance. To test the degree to which we could control network connectance, we also included a combination treatment, with both fertilizer and herbicide, that we expected not to differ from the control. Our hypotheses were: (1) fertilization would increase floral resource quality in a way that promoted a higher insect abundance and species richness, resulting in increased network connectance, (2) herbicide exposure would have the opposite effect, reducing visitor abundance and species richness through a decrease in floral resource quality, resulting in decreased network connectance, (3) the combination of both herbicide and fertilizer exposure would not differ significantly from the control because increases and decreases in attractiveness would result in a neutral effect on insect abundance and species richness.

## MATERIALS AND METHODS

We conducted the field experiment in Dublin, Ireland in 2017 and 2018. In 2017, we planted four 2 × 2 m experimental research plots at four separate sites (total 16 plots). At each site, the plots were separated by more than 50 meters, and the sites were separated by at least 1 km. In each of the plots, we planted the same controlled plant community, including six native perennial species (*Cirsium vulgare* Ten. and *Hypochaeris radicata* L. Asteraceae, *Epilobium hirsutum* L. Onagraceae, *Filipendula ulmaria* Maxim. Rosaceae, *Plantago lanceolata* L. Plantaginaceae, and *Oregano vulgare* L. Lamiaceae) and one non-native annual species (*Phacelia tanacetifolia* Benth. Boraginaceae) in the same densities in every plot (Supplemental Material). The four plots at each site were randomly assigned one of the four experimental treatments: low concentrations of fertilizer exposure (F), low concentrations of herbicide exposure (H), low concentrations of both fertilizer and herbicide (HF), and a control of just water (C) (Table 1, Supplemental Material). In 2018, we replicated the same experiment at four new sites. Thus, the experiment had a replicated block design, with the eight sites as blocks (*Russo et al., 2020*). All study sites were separated by at least 1 km. A map of study sites and detailed description of each site is provided in the Supplemental Material (Figs. S1, S2).

The low concentration agrochemical treatments were applied in 10 L water once a week for 3 months during the flowering period. The highest concentrations were applied in the first month of the season, and the concentrations decreased each month, with the lowest concentrations applied in the third month. We applied decreasing concentrations of fertilizer and herbicide through the summer to simulate field-realistic non-target exposure scenarios, given a likely spring application of fertilizer and herbicide in a typical agroecosystem (*Bertol et al., 2007*; *Craig & Mannix, 2009*; *Korsaeth & Eltun, 2000*). Control plots received the same volume of untreated water as the treatment plots on the same days. The total herbicide exposure (summed across the season) was approximately equivalent to

**Table 1 Concentrations of fertilizer and herbicide applied at treatment plots over time, as well as total annual application.**

|  | First month (once a week) | Second month (once a week) | Third month (once a week) | Total annual application |
|---|---|---|---|---|
| N (mg/l) | 30 | 20 | 10 | 0.6 g/m$^2$ |
| P (mg/l) | 15 | 10 | 5 | 0.3 g/m$^2$ |
| K (mg/l) | 5.5 | 3 | 1 | 0.095 g/m$^2$ |
| Glyphosate (mg active ingredient/l) | 0.7 | 0.3 | 0.1 | 0.011 g/m$^2$ |

7.6% of a standard annual field application of 1,440 g/ha (*Dupont, Strandberg & Damgaard, 2018*) and the highest concentration applied was less than half the maximum level detected in a large-scale European soil survey (*Silva et al., 2019*).

Previous work showed that these agrochemical treatments affected plant growth in the following ways. On average, plants exposed to the low concentration of fertilizer in our study flowered earlier in the year than plants exposed to a control of just water (*Russo et al., 2020*). Plants exposed to the low concentration of herbicide in our study were shorter and had shorter leaves than control plants exposed to just water (*Russo et al., 2020*). However, there was no effect of the agrochemical treatments on the size of the floral display (number of inflorescences multiplied by average inflorescence size) (*Russo et al., 2020*). Other work demonstrated that plants exposed to low concentrations of herbicide had lower total amino acid concentrations, and plants exposed to low concentrations of fertilizer lower total fatty acid concentrations in their pollen (*Russo et al., 2023*). That study also showed that plants exposed to fertilizer had higher total quantities of pollen per flower (*Russo et al., 2023*). Thus, the experimental treatments had effects on the quality of floral resources, but not on the size of the floral display.

Over the course of 2017 and 2018, we conducted standardized 5-min floral visitor sampling periods on each plant species in bloom, in each plot, once a week throughout the flowering season. The first two plant species, *P. tanacetifolia* and *P. lanceolata*, began to bloom in late April, followed by *H. radicata* in early May, then *F. ulmaria*, *E. hirsutum*, and *C. vulgare* in late May and *O. vulgare* at the beginning of June, and bloom continued through August in both years. All plant species in the study overlapped in their bloom from early June through the end of the flowering season (10 weeks, Fig. S3). Insects contacting the reproductive parts of the inflorescences during the sampling period were collected and stored in a −20 C freezer until the end of the flowering season, when they were pinned, labeled, and sorted to morphospecies. Species identifications were verified by qualified taxonomists at the National Biodiversity Data Centre (bees) and Martin Speight (hoverflies). Species are vouchered at Trinity College Dublin, Dublin, Ireland.

We conducted a rarefaction analysis to test our sample coverage for the insect specimens and to compare three Hill number diversity indices among the four experimental treatments (*Chao et al., 2014*). For this rarefaction analysis, we used the *iNEXT* package in R (*Hsieh, Ma & Chao, 2016*).

During each sampling event, we counted the number of inflorescences of each plant species in bloom. We also measured 20 randomly selected inflorescences of each species and measured the total floral area of the inflorescences. We calculated floral display for each sampling event using the total number of inflorescences multiplied by the average inflorescence area.

## Data analysis

All analyses were conducted using the software R (*R Core Team, 2022*). For the purposes of this study, our focus was on the effects of the experimental manipulations on network structure. We tested the degree to which we could both *increase, decrease*, and *maintain* connectance using chemical fertilizers and herbicides.

Our first hypothesis was that fertilization would increase floral resource quality in a way that promoted a higher insect abundance and species richness (*Russo et al., 2023*). By this mechanism, fertilizer would experimentally increase network connectance. We also hypothesized that herbicide exposure would have the opposite effect, reducing visitor abundance and species richness through a decrease in floral resource quality. Finally, we hypothesized that the combination treatment of both herbicide and fertilizer exposure would not differ significantly from the control because increases and decreases in attractiveness would result in a neutral effect on insect abundance and species richness.

To quantify the effect of our experimental manipulation on network connectance, we constructed weighted bipartite networks of the season-aggregated plant-visitor data for each plot. For each plot network, we calculated total insect abundance and species richness, unweighted degree, weighted degree, nestedness (NODF), NODFc, modularity, and connectance (the realized proportion of all potential interactions). The unweighted degree was calculated as the total number of unique interactions divided by the number of species in each network. The weighted degree was calculated as the total abundance of visitors divided by the number of species in the network. The connectance was calculated as the number of unique interactions (*i.e.*, unweighted degree) divided by the number of plant species that bloomed in each plot multiplied by the total number of possible insect visitor species (experiment-wide insect species richness). The nestedness was calculated in two ways: first, we used the "NODF" function in the R package *bipartite* (*Dormann, Gruber & Fründ, 2008*) and second, we used the "NODFc" function in the R package *maxnodf* (*Song, Rohr & Saavedra, 2017*; *Hoeppke & Simmons, 2021*). NODFc is calculated relative to network size for nestedness analyses (*Hoeppke & Simmons, 2021*). Modularity was calculated using the "computeModules" function in the R package *bipartite*. We tested for correlations between all of these variables.

We used generalized linear mixed effects models (GLMMs, package *lme4* in R) to test whether our experimental treatments had significant effects on all of these network measures (*Bates et al., 2014*). For these models, the fixed effects were experimental treatment and the log-normalized floral display and the random effect was the experimental block (n = 8, four sites in 2017 and four in 2018). We tested for significant interactions between floral display and treatment for these models, and excluded the interaction term where it was not significant.

To determine whether common generalist floral visitors visited a different number of plant species in the different experimental treatments, we used a Kruskal-Wallis one-way analysis of variance. We individually tested the top ten most abundant floral visitor species to see whether the number of plant species visited within the treatment plots differed from the control, using a Bonferroni correction for multiple testing (package *stats*, *R Core Team, 2022*).

We used a null model analysis to compare the network measures among the different treatments and between these treatments and randomized networks. We generated 500 null models for each of the 31 networks using the *nullmodel* function (package *bipartite* in R), with the "r2dtable" method, which randomizes interactions while maintaining marginal totals (*Dormann, Gruber & Fründ, 2008*). Next, we measured six network measures (connectance, weighted degree, unweighted degree, modularity, nestedness (NODF), and NODFc) for all of the null models. We calculated z scores for the observed network measures relative to the mean and standard deviation of the null model values. We then used GLMMs (package *lme4*) to test whether the z-scores of the agrochemical treatment networks differed significantly from the z-scores of the control. We also report which treatment networks differ significantly from the null models as those where the mean ± the standard deviation of the z-scores did not overlap with 0.

## RESULTS

We collected 1,908 insect specimens representing 92 species or morphospecies during ~117 h of sampling across the eight sites in the 2 years (Table S1). Across all the plots, only one failed to receive enough visitors to be included in the analysis (control plot at site 5 in 2018). A rarefaction analysis indicated that our sample coverage across the experimental treatments was over 96% (Fig. S4). Thus, we were able to construct 31 plot level networks distributed evenly across the eight blocks, with the exception of the one control plot (Supplemental Material).

We tested for correlations between all measured variables and found that, aside from modularity and nestedness, most variables included here exhibited significant positive correlations with connectance (Fig. S5).

The log-normalized size of the floral display was positively associated with floral visitor abundance and species richness, along with weighted and unweighted degree, modularity, connectance and NODFc (Table 2, Fig. 1). In addition, floral display interacted significantly with the herbicide treatment for NODF (Fig. S6). However, our rarefaction analysis indicated that the treatments overlapped in the diversity of insect species visiting the plants, as measured by the three Hill numbers (Fig. S4).

Though there was no treatment effect on the size of the floral display, the experimental treatments significantly affected both the abundance and species richness of floral visitors in the plots (Table 2), which led to significant changes in connectance (Fig. 2).

Explicitly, herbicide exposure decreased floral visitor abundance, and fertilizer exposure increased floral visitor species richness. Both the visitation frequency (abundance) and species richness correlated with connectance, but only the fertilizer treatment had a

**Table 2 Model structure and results from the generalized linear mixed effects models.**

| Response | Random effect | Observations | Fixed effect | Contrasts | Effect size | T value | P value | R²m | R²c |
|---|---|---|---|---|---|---|---|---|---|
| log (Floral display) | Block | 31 obs, 8 blocks | Treatment | C—F | 0.06 | 0.21 | 0.84 | 0.03 | 0.43 |
| | | | | C—H | −0.28 | −0.91 | 0.36 | | |
| | | | | C—HF | −0.08 | −0.27 | 0.79 | | |
| log (Abundance) | Block | 31 obs, 8 blocks | **Treatment** | C—F | 0.16 | 1.23 | 0.22 | 0.76 | 0.93 |
| | | | | **C—H** | **−0.33** | **−2.5** | **0.01** | | |
| | | | | C—HF | 0.004 | 0.03 | 0.98 | | |
| | | | | **log (Display)** | **1.02** | **11.45** | **<0.001** | | |
| Floral visitor species richness | Block | 31 obs, 8 blocks | **Treatment** | **C—F** | **4.68** | **3.88** | **<0.001** | 0.48 | 0.88 |
| | | | | C—H | −0.23 | −0.19 | 0.85 | | |
| | | | | C—HF | 1.92 | 1.59 | 0.11 | | |
| | | | | **log (Display)** | **5** | **6.05** | **<0.001** | | |
| Weighted degree | Block | 31 obs, 8 blocks | Treatment | C—F | −0.16 | −0.61 | 0.55 | 0.74 | 0.89 |
| | | | | C—H | −0.3 | −1.11 | 0.27 | | |
| | | | | C—HF | 0.16 | 0.61 | 0.54 | | |
| | | | | **log (Display)** | **1.69** | **9.72** | **<0.001** | | |
| Unweighted degree | Block | 31 obs, 8 blocks | Treatment | C—F | 0.04 | 0.45 | 0.65 | 0.36 | 0.6 |
| | | | | C—H | −0.09 | −1.14 | 0.26 | | |
| | | | | C—HF | −0.07 | −0.92 | 0.36 | | |
| | | | | **log (Display)** | **0.16** | **3.4** | **<0.001** | | |
| Connectance | Block | 31 obs, 8 blocks | **Treatment** | **C—F** | **0.01** | **2.59** | **0.009** | 0.32 | 0.77 |
| | | | | C—H | 0.002 | 0.4 | 0.69 | | |
| | | | | C—HF | 0.004 | 0.81 | 0.42 | | |
| | | | | **log (Display)** | **0.01** | **3.72** | **0.0002** | | |
| Nestedness (NODF) | Block | 31 obs, 8 blocks | **Treatment** | C—F | −149.2 | −1.55 | 0.12 | 0.52 | 0.55 |
| | | | | **C—H** | **−166.08** | **−2.14** | **0.03** | | |
| | | | | C—HF | −25.21 | −0.35 | 0.73 | | |
| | | | | log (Display) | 3.5 | 0.56 | 0.58 | | |
| | | | **Interaction** | F:Display | 14.18 | 1.51 | 0.13 | | |
| | | | | **H:Display** | **16.48** | **2.16** | **0.03** | | |
| | | | | HF:Display | 1.44 | 0.2 | 0.84 | | |
| NODFc | | | Treatment | C—F | −0.14 | −0.68 | 0.50 | 0.28 | 0.28 |
| | | | | C—H | 0.05 | 0.21 | 0.84 | | |
| | | | | C—HF | −0.37 | −1.77 | 0.09 | | |
| | | | | **log (Display)** | **0.25** | **2.49** | **0.02** | | |
| Modularity | Block | 31 obs, 8 blocks | Treatment | C—F | 0.04 | 1.23 | 0.22 | 0.52 | 0.67 |
| | | | | C—H | 0.003 | 0.09 | 0.93 | | |
| | | | | C—HF | 0.06 | 1.8 | 0.07 | | |
| | | | | **log (Display)** | **−0.1** | **−5.15** | **<0.001** | | |

Note:
Model structure and results from the generalized linear mixed effects models. The model structure includes the response variable (Response), random effect, and fixed effects. We report the number of observations and groups in the random effect, as well as the individual contrasts that we tested in each model. We report the effect size, t value, and p value from each contrast. We also report the marginal (R²m) and conditional (R²c) of each model. The marginal R² expresses the percent variance in the response variation explained by the fixed effects in the model, while the conditional R² expresses the percent variance in the response variable explained by both the fixed and random effects in the model. Significant contrasts appear in bold.

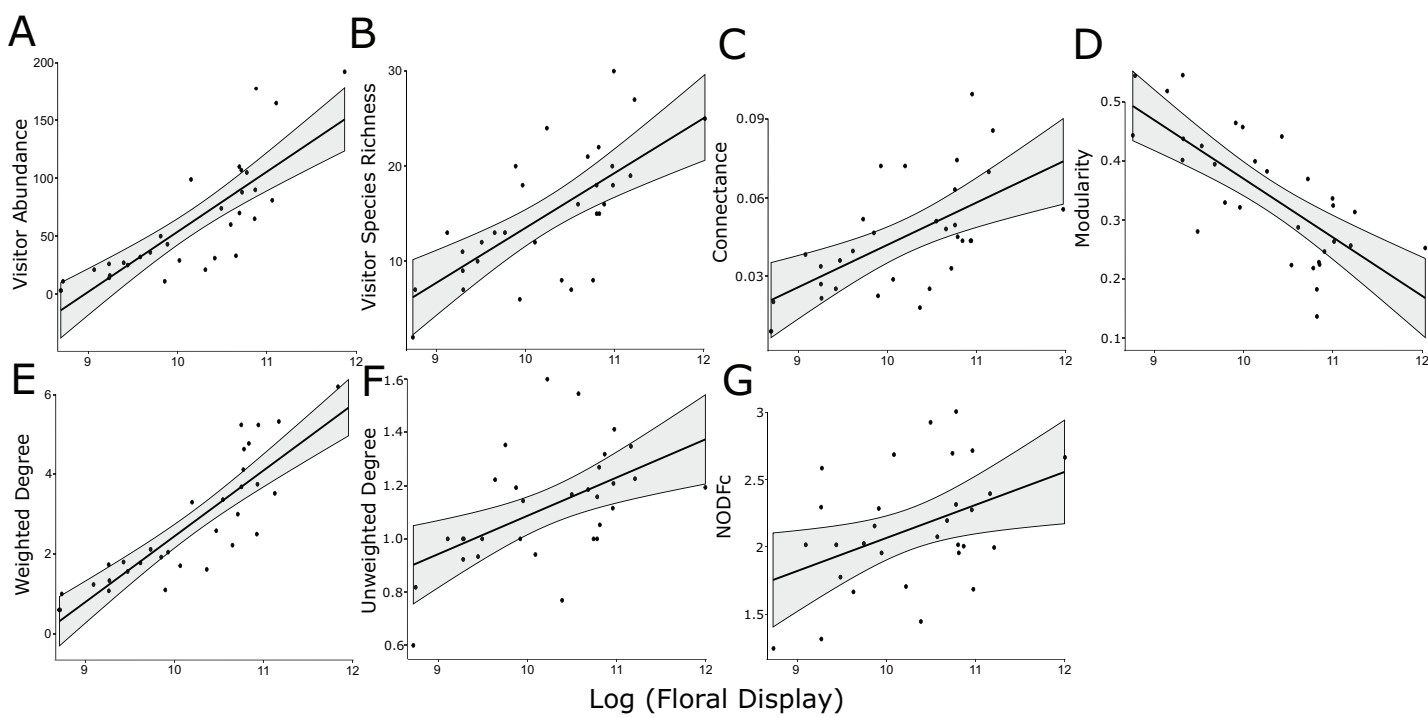

**Figure 1 Significant linear relationships between the log-transformed size of the floral display and floral visitor abundance.** (A) Floral visitor species richness, (B) network connectance, (C) modularity, (D) weighted degree, (E) unweighted degree, (F) and NODFc (G).

connectance that differed significantly from the control (Fig. 3). All measured attributes of the 31 networks are provided in the Supplemental Material (Table S3).

Among the top 10 most abundant flower-visiting insects in the experiment (*Apis mellifera*, *Bombus lapidarius*, *Bombus lucorum* agg., *Bombus pascuorum*, *Bombus pratorum*, *Bombus terrestris*, *Episyrphus balteatus*, *Platycheirus albimanus*, *Sphaerophoria scripta*, and *Syrphus ribesii*), we only detected significant changes in the number of plant species visited by *A. mellifera* in the treatment plots. *A. mellifera* visited significantly more plant species in fertilized plots (mean rank difference = 35.86, *p* = 0.01).

Our null model analysis showed that only the herbicide exposed networks differed significantly from the z-scores of the control networks (Table S4). The herbicide networks had significantly higher (in this case, closer to 0) z-score for connectance, NODF, and unweighted degree. In all these cases, the herbicide networks did not differ from the null models; herbicide networks only differed from the null models in terms of modularity (Fig. S7). The other treatments differed significantly from the null modes in all cases except that all but the combination network did not differ in NODFc, the control network did not differ in unweighted degree, and the combination treatment did not differ in weighted degree (Fig. S7). We also provide visualizations of the aggregated treatment networks in the Supplemental Materials (Fig. S8).

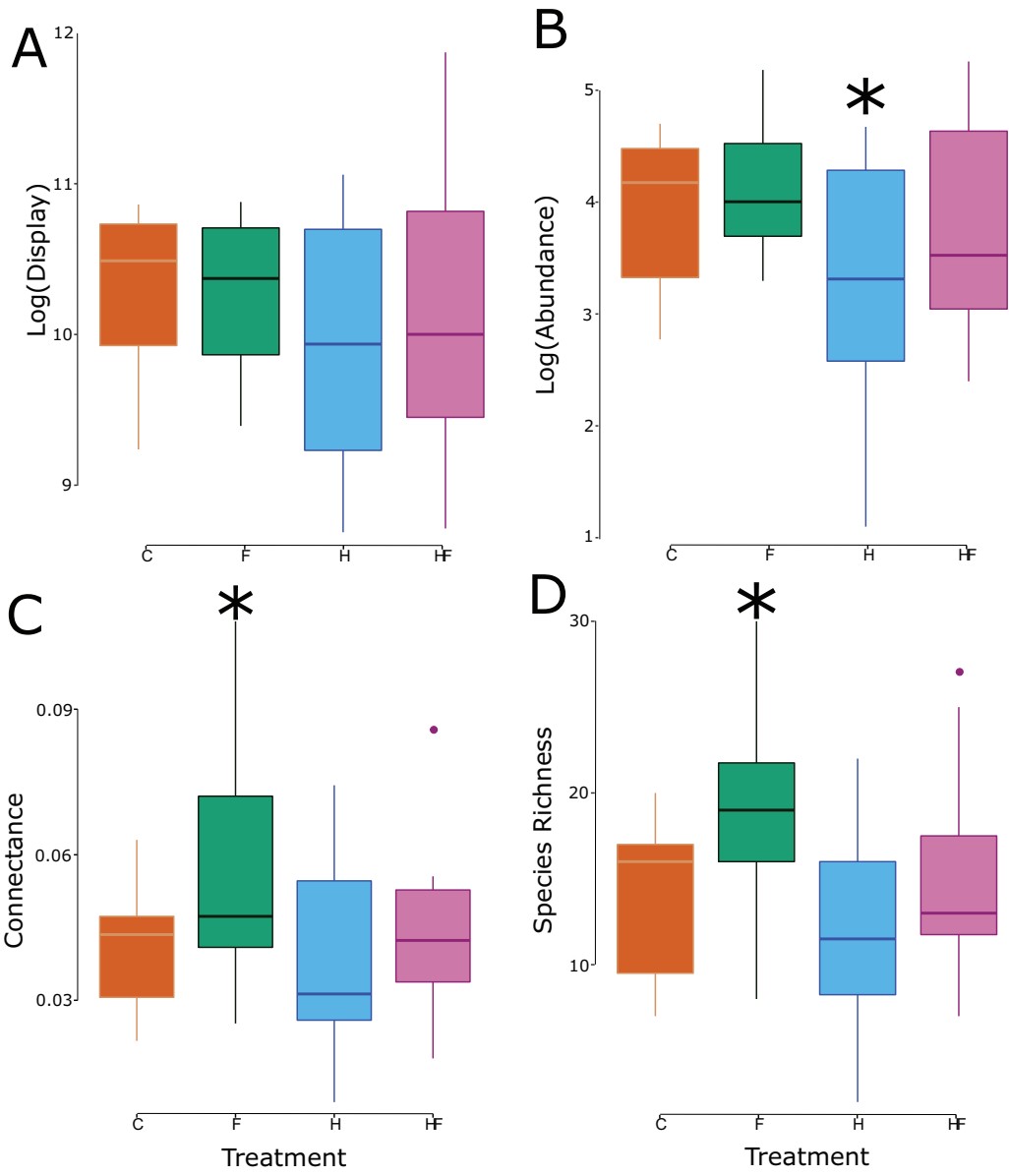

**Figure 2 Box and whisker plots of the relationship between the experimental treatment (red = control, green = fertilizer, blue = herbicide, purple = combination) and the log-transformed floral display (A), log-transformed floral visitor abundance (B), network connectance (C), and floral visitor species richness (D). Significant effects (relative to the control treatment) are marked with an asterisk.**

## DISCUSSION

In this study, we addressed three main hypotheses, the first of which was that network connectance would increase in the fertilized treatment due to higher insect abundance and species richness because of increased floral resource quality. We provided support for the first hypothesis; we were able to successfully increase network connectance with a fertilizer treatment. We observed this increase in connectance despite the lack of a significant increase in floral display, suggesting that the attractiveness, rather than the abundance, of

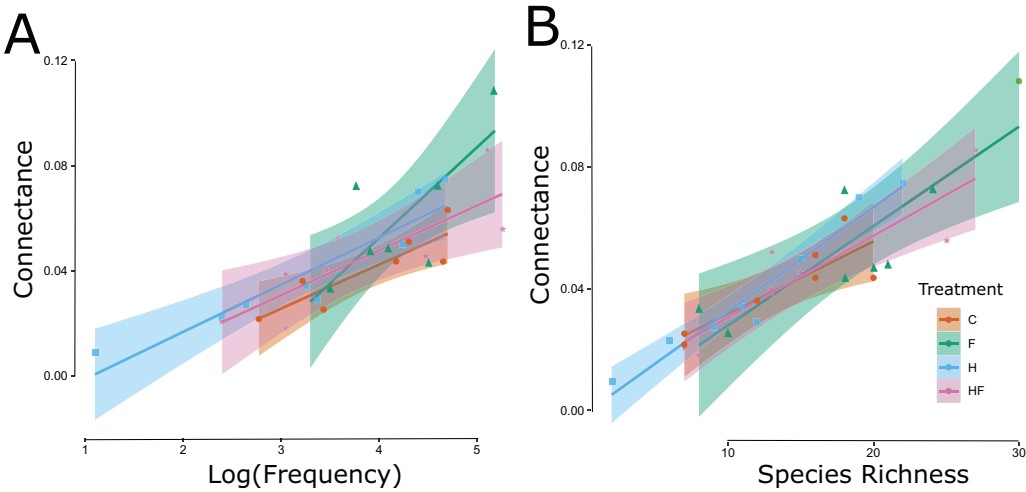

**Figure 3 Significant linear relationships between the log-transformed frequency of visitation (floral visitor abundance) (A) and floral visitor species richness (B) and the network connectance.** The experimental treatments are indicated by color (red = control, green = fertilizer, blue = herbicide, purple = combination) and symbol type (circle = control, triangle = fertilizer, square = herbicide, diamond = combination).

the flowers drove the increase (*Russo et al., 2023*). Such changes in floral resource quality, and the attractiveness to flower-visiting insects, have been demonstrated by other fertilization studies (*Cardoza, Harris & Grozinger, 2012*). Moreover, this increase in connectance was significantly related to an increase in the species richness of flower-visiting insects in our experimental plots. This result agrees with other work showing increases in insect visitation and abundance on plants exposed to fertilizer (*Burkle & Irwin, 2009*; *Dupont, Strandberg & Damgaard, 2018*), and supports a previous study showing that it is possible to experimentally manipulate network connectance using chemical fertilizers (*Russo & Shea, 2016*).

On the other hand, we did not provide support for our second hypothesis that herbicide exposure would decrease network connectance due to reduce visitor abundance and species richness because of a decrease in floral resource quality. We did not see a corresponding significant decrease in network connectance associated with an herbicide treatment. Plants exposed to herbicide received a significantly decreased abundance of flower-visitors in our study, in agreement with other studies of the effects of low concentrations of herbicide exposure on flower visitors (*Dupont, Strandberg & Damgaard, 2018*). Though floral visitor abundance correlated with connectance, this decrease did not result in a corresponding significant change in the connectance. It seems in this case that changes in connectance were driven by the species richness of the visitors. Here, plants exposed to low concentrations of herbicide did not have a lower floral visitor species richness than the control plants.

Finally, we found support for our third hypothesis that herbicide and fertilizer exposure together would not differ significantly from the control. We did not see significant differences between the control (water) and combination (herbicide and fertilizer) treatments. This supports our hypothesis that the contrasting effects of the fertilizer and

herbicide resulted in balanced effects for insect visitation and agrees with a previous study that evaluated the interaction between herbicide and nitrogen fertilizer on plant traits and visitation patterns (*Dupont, Strandberg & Damgaard, 2018*). Thus, our targeted treatments to *increase* (fertilizer) and *maintain* (combination) connectance were successful, while our treatment to *decrease* (herbicide) connectance was not.

When investigating changes in network connectance, it is important to distinguish between effects that may relate to a conservation objective, such as maintaining biodiversity, *vs* changes in the preferences of already common visitor species. In our study, of the ten most abundant floral visitor species in our study, only honey bees (*Apis mellifera*) visited a greater number of plant species in fertilized plots. This suggests that the increase in connectance was a direct result of increased species richness of flower-visitors in the fertilized plots, rather than an increase in the connectance of already generalist flower-visitors. In other words, increased connectance was due to the plants attracting a greater proportion of possible flower visitors in the network, rather than common flower visitors increasing the proportion of possible plants they visited. Although the relationship between connectance and conservation value has not been clearly established (*Heleno, Devoto & Pocock, 2012*), this increase in the species richness of flower visitors is encouraging for systems where supporting diverse populations of wild insects is a management goal.

Our methodology did not allow us to determine whether the insects visiting flowers were effecting pollination. Determining per-visit pollination efficacy (*e.g.*, *Ne'eman et al., 2010*) is time-consuming and would have limited the amount of data we were able to collect for our community analysis. However, there are fitness implications for studying flower visitors rather than pollinators in the strict sense (*King, Ballantyne & Willmer, 2013*). Because true pollination networks are significantly more specialized than flower-visitor networks (*King, Ballantyne & Willmer, 2013*), future work could explore whether network connectance changes when only true pollinators are considered. Such a step would provide important insight into the relationship between changes in connectance and network function (*Popic, Wardle & Davila, 2013*).

In our study, we used very low concentrations of fertilizer and herbicide. Higher concentrations of herbicide may have led to a reduction in connectance, but may also have caused increases in mortality that would confound the results. Other research has shown that other stressors, including temperature and water stress, may reduce the quality of floral resources provided by plants (*Mu et al., 2015*; *Phillips et al., 2018*; *Rering et al., 2020*; *Descamps et al., 2021*) and we may therefore hypothesize that such stressors have the potential to decrease network connectance. We also controlled plant community composition in our study, whereas in a field-realistic scenario, the plant community composition has been demonstrated to change with agrochemical exposure (*Damgaard et al., 2022*). Moreover, these agrochemicals may directly affect the health of flower-visiting insects, resulting in less predictable longer-term effects on visitation patterns (*Zioga, White & Stout, 2022*). Further research may explore the effects of such longer-term changes on network connectance.

## CONCLUSIONS

Networks of mutualistic interactions represent the architecture of biodiversity and produce critical ecosystem services, such as pollination. These complex sets of interactions produce predictable patterns, and we have shown that we can manipulate these patterns in predictable ways, successfully increasing and maintaining network connectance with agrochemical treatments. Work evaluating the structural controllability of plant-pollinator networks has demonstrated that controllability relates strongly to stability and conservation or management objectives (*Cagua, Wootton & Stouffer, 2019*). Our work shows that, where network measures are a conservation objective (*e.g.*, *Tylianakis et al., 2010*), they can be managed.

## ACKNOWLEDGEMENTS

We thank the sites that allowed us to use their land for our study, and access their water taps: University College Dublin and the Lamb Clarke Irish Historical Apple Collection at Rosemount Environmental Research Station, Gas Networks Ireland, Raidió Teilifís Éireann, Trinity College Dublin, Marino Institute, Riverview Educate Together National School, and Airfield Estate. Special thanks to Dr. W. Deasy, B. Moran, Dr. K. McAdoo, T. Bannon, C. van der Kamp, R. Hession, C. Bennett, E. Kavanagh, C. Fogarty, S. Austin, M. Burke, R. Judge, S. Waldren, E. Bird, and M. McCann for technical assistance, S. Palumbo, A. Flaherty, J. Stone, S. McNamee, B. Malone for field help, and O. Fenton, D. OHuallachain, J. Finn, J. Zimmerman, J. Parnell, and S. Hodge for advice. Species identifications of the bees were verified at the National Biodiversity Data Centre in Waterford, Ireland with help from Ú. Fitzpatrick and T. Murray, while syrphid fly identifications were corrected and validated by M. Speight, of Trinity College Dublin. For insects other than syrphid flies and bees, we received additional assistance from specialists M. Smith and B. Nelson (National Parks and Wildlife Service).

### Funding

Funding for this study was provided by a Marie Curie Independent Fellowship [grant number FOMN-705287] to LR and JS. The funders had no role in study design, data collection and analysis, decision to publish, or preparation of the manuscript.

### Grant Disclosures

The following grant information was disclosed by the authors:
Marie Curie Independent Fellowship: FOMN-705287.

### Competing Interests

The authors declare that they have no competing interests.

## Author Contributions

- Laura Russo conceived and designed the experiments, performed the experiments, analyzed the data, prepared figures and/or tables, authored or reviewed drafts of the article, and approved the final draft.
- Jane C. Stout conceived and designed the experiments, authored or reviewed drafts of the article, and approved the final draft.

## Data Availability

The raw data are in the Supplemental Files.

## Supplemental Information

Supplemental information for this article can be found online at http://dx.doi.org/10.7717/peerj.16319#supplemental-information.

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
