# Peer review of "Manipulating network connectance by altering plant attractiveness"

_PeerJ, doi:10.7717/peerj.16319_

## Round 0.1 · original submission · Major Revisions

Dear Dr. Russo,

After this first review round, three reviewers evaluated your manuscript, which received a major and two minor reviews. Considering all issues raised by the reviewers, I believe your manuscript needs a major review before it is accepted for publication in PeerJ.

Therefore, I invite you and your co-authors to assess the reviews and prepare a new version of your text. Please do not forget to prepare a rebuttal letter informing the reviewers what changes were made and why or explain why there were no changes in specific criticisms. Finally, do no hesitate to write me in case you have any doubts or to resubmit earlier if you can.

Best regards,
Daniel Silva

Reviewer 1 ·

Basic reporting

This is an interesting study by Russo & Stout, where they analyze the effect of chemical fertilizers and herbicides on plant pollinator networks. They extrapolate the practical implications of this approach to the ability to experimentally manipulate network connectance. Overall, the text is well written. The English used throughout the text is clear and professional. The organization of the paper, figures, and tables are appropriate. Background information is sufficient and the paper includes a reasonable number of references. Relevant results are presented and raw data are shared. I only propose to make the hypothesis explicit in the last paragraph of the Introduction, since it first appeared in the Materials and Methods (L 123-130).

Experimental design

The experimental design is well thought out and in line with the aims and scope of the journal. The use of 2x2 plots with planted plant species allows for standardized comparisons between treatment blocks and is easily replicable. The research questions are well-defined, relevant, and meaningful. The study fills a knowledge gap by providing experimental evidence on the drivers of plant-pollinator network patterns. The investigation is conducted rigorously with high technical and ethical standards.

However, I have some concerns:
1. Why the concentrations of fertilizer and herbicide applied to the treatment plots decreased over the sampling period. I miss a clear explanation of how the very low concentrations in the herbicide treatment can be compared to the control treatment. The authors should address this in the discussion.
2. Were all plant species in flower throughout the sampling period? I suggest presenting the flowering period of all plant species to account for its influence on floral display and pollination attractiveness.
3. I noticed that measuring pollination effectiveness as "insects contacting reproductive parts of inflorescences" does not clearly distinguish between visitors and pollinators. This problem has been well documented in previous studies (Fishbein & Venable 1996; Ne'eman et al. 2010), including specifically in the case of network studies (see Popic et al. 2013; King et al. 2013). I then suggest acknowledging this limitation in the discussion and using the term "floral visitors" instead of "pollinators".

References:
Fishbein, M., Venable, D. L., 1996. Diversity and temporal change in the effective pollinators of Asclepias tuberosa. Ecol., 77(4), 1061-1073.
Ne'eman, G., Jürgens, A., Newstrom‐Lloyd, L., Potts, S. G., Dafni, A., 2010. A framework for comparing pollinator performance: effectiveness and efficiency. Biol. Rev., 85(3), 435-451.
Popic, T. J., Wardle, G. M., Davila, Y. C., 2013. Flower‐visitor networks only partially predict the function of pollen transport by bees. Austral Ecol., 38(1), 76-86.
King, C., Ballantyne, G., Willmer, P. G., 2013. Why flower visitation is a poor proxy for pollination: measuring single‐visit pollen deposition, with implications for pollination networks and conservation. Methods Ecol.Evol., 4(9), 811-818.

Validity of the findings

The reviewer points out that the conclusions are only partially stated. The authors discuss the practical implications of their approach to manipulating network connectance, but do not explicitly answer their research question or support/reject their hypothesis. The limitations of the study are acknowledged and meaningful replications are suggested for future studies, such as testing higher concentrations of herbicide and assessing longer-term effects on network connectance. The reviewer finds the data analysis to be robust, statistically sound, and well controlled, and appreciates the provision of raw data. Overall, my assessment is generally positive, with some specific concerns and suggestions for improvement regarding experimental design and clarity of conclusions.

Reviewer 2 ·

Basic reporting

I have read with interest the manuscript entitled “Manipulating network connectance by altering plant attractiveness”, by Russo and Stout. The authors aimed to experiment with connectance, a structural attribute reflecting insect visitation patterns, in ecological networks. They conducted a two-year field experiment in Dublin, applying four agrochemical treatments to fixed communities of seven flowering plant species. They collected 1,908 flower-visiting insects of 92 species using standardized sampling methods.
The authors hypothesize that the fertilizer treatment would increase connectance, the herbicide treatment would decrease it, and a combination of both treatments would maintain it compared to a water-only control. The authors report a successful connectance increase with the fertilizer treatment and maintenance with a combination of fertilizer and herbicide. However, the herbicide treatment did not decrease connectance. The rise in connectance in the fertilized treatment resulted from increased species richness of visiting insects, not their abundance, particularly an increase in the proportion of insect visitor species. They conclude that connectance can be manipulated in ecological networks, with implications for management and conservation efforts in mutualistic communities.
I have however some questions regarding this manuscript that prevents me from recommending its publication in the present form. These are detailed below, and I hope will help in improving a future version of the manuscript. You will find the suggestions in this and the following sections of the review.

-In terms of basic reporting, the use of connectance to support the importance of the study should be reinforced in both the introduction and discussion sections.

High connectance is often associated with pristine or near-pristine communities, protecting them from secondary extinctions. It is implied that highly connected communities are desirable from a conservationist standpoint, assuming a positive relationship between connectance and conservation value. However, the prevalence of this relationship has been heavily questioned (e.g., Tylianakis et al., 2010). For instance, Heleno et al. (2012) conducted a literature review to search for empirical evidence of a relationship between connectance (complexity) and conservation value in communities at different stages of degradation. They demonstrated that the often assumed positive relationship between highly connected and desirable communities (i.e., with high conservation value) does not derive from empirical data, and this topic deserves further discussion. Consequently, these authors suggest that connectance on its own cannot provide clear information about conservation value. Please, mention this debate in the introduction and the discussion."

Although both references are cited in the manuscript, strangely their purpose of questioning connectance is neither mentioned nor suggested in the current version.

Tylianakis, J. M., Laliberté, E., Nielsen, A., & Bascompte, J. (2010). Conservation of species interaction networks. Biological conservation, 143(10), 2270-2279.

Heleno, R., Devoto, M., & Pocock, M. (2012). Connectance of species interaction networks and conservation value: is it any good to be well connected?. Ecological indicators, 14(1), 7-10.

Experimental design

The methods are mostly rigorous but some parts still need to be clearly explained and with sufficient detail. Many of the samples are impressively large and are well replicated and controlled. However, as I mentioned before, there are methodological parts that need further explanation.

- Please provide more details about the dates when your study was conducted. For instance, which months did it take place?

- In the same vein as the previous question, was the flowering of all the plants used in the quadrants synchronous? Did it have the same duration? These are essential details as they relate to insect visitation.

-The authors mention that in 2018, they replicated the same experiment at 4 new sites. Please, specify where these sites were located. Were they in the same region? What was the distance between them?

-It is necessary for the authors to provide a precise description of the sites where they established their experimental plots, as it will undoubtedly help contextualize their results. For instance, what type of vegetation is present in the surroundings? Were there other species of wild plants attractive to insects with flowering during the observations?

-One relevant piece of information, given the use of fertilizers, herbicides, and artificial irrigation for the study, is to mention the type of soil in the studied sites. Was it the same?

-Is there any commercial name for the fertilizer and herbicide used? Please provide them.

-An herbicide is a chemical product used to control/eliminate unwanted plants. Some act by interfering with the growth of specific plants, while others have a broader effect on all present species. Please, mention the effect it had on the observed plants in your experiments.

-The authors focused on the effects of experimental manipulations on the structure of plant-insect interaction networks. To do this, in addition to connectance, they used metrics such as nestedness (unweighted and weighted) and modularity. They mention having calculated these values for 31 interaction networks. However, neither in the text nor in the supplementary files can the values of these metrics be seen (The figures are not very helpful for this purpose). Please, as these values are fundamental for their study, they should be shown.

-The authors also do not mention the use of null models for the estimation of the analyzed network metrics. Null models can help to determine if observed network structure is significantly different from random network structure. Also, the use of null models is necessary to compare structural properties between networks (as in your study). Common solution is to calculate the z-score to standardize metrics first, then compare the z-scores of a metric between networks. Please, present these analyses to validate your obtained results.

- The sampling of pairwise interactions between species demands even greater effort due to the total number of interactions being higher than the total number of species, and the difficulty in observing these interactions. Consequently, reporting the sampling completeness of interactions is essential. Sampling completeness is a relative measure that enables a comparison of the degree of under-sampling between networks that vary in size (as in your study). The authors must report this information, for example, by using the Chao family of asymptotic species richness estimators to predict the total number of interactions (Chao & Jost 2012).

Chao A, Jost L. 2012. Coverage-based rarefaction and extrapolation: Standardizing samples by completeness rather than size. Ecology 93 (12): 2533–2547.

- The authors report obtaining 31 interaction networks. However, the values of the metrics for each of these networks are not shown. In a supplementary file, network plots are displayed for each of the treatments (i.e., 3 networks). The authors do not specify whether these networks are an average of those obtained throughout the study. This should be clarified.

Validity of the findings

This manuscript has a logical structure. The results are interesting and based on a robust data set. However, throughout the manuscript, there are several gaps in information that need to be addressed to clarify the study, and all of them were mentioned above.

Additional comments

No comments

Reviewer 3 ·

Basic reporting

No comment

Experimental design

The experimental design is appropriate and the methodology is clear. The only exception is that it was not clear whether some of the metrics used were computed using the weighted or unweighted network. For degree this is clear, however, for connectance, nestedness and modularity I could not be sure. Furthermore, I could not find in the text how these metrics were computed (e.g., what software or package was used).

Validity of the findings

No comment

---

## Round 0.2 · Minor Revisions

Dear Dr. Lara,

After this new review round, one reviewer accepted your manuscript, while the other indicated minor reviews are required. Specifically, the reviewer suggested the application of another metric. Please consider applying this new metric or justify the reason why there is no need to apply it.

Please note that my field of expertise is not network analysis. So, I will need to rely on the second review from that reviewer to reach the final decision on your study.

Sincerely,
Daniel Silva

Reviewer 1 ·

Basic reporting

The authors have satisfactorily addressed all the concerns and queries I raised during the first round of revision. The paper has seen significant improvements, particularly through the inclusion of additional data regarding phonological flowering overlap among the studied plant species. Additionally, the authors have appropriately acknowledged the limitations of their study, acknowledging that they focused on a plant-floral visitor network rather than a true plant-pollinator network. This awareness of limitations adds depth to the study and opens up avenues for future research in the field of plant-pollinator networks.

Experimental design

no comment

Validity of the findings

no comment

Additional comments

Considering the above improvements, I recommend the publication of the manuscript in its current form. I eagerly anticipate the publication of this paper, as I believe it will make a valuable contribution to our understanding of the impacts of increased chemical fertilizer use and decreased chemical herbicide application on urban plant-pollinator networks.

Reviewer 2 ·

Basic reporting

I have read the new version of the manuscript with great pleasure. I have verified that all of my suggested changes were properly addressed by the authors. I am very grateful for the effort that has been put into it, resulting in a more robust and clear version of the manuscript. Nevertheless, I do have one final recommendation, which I believe the authors can address without difficulty.
There are well-known issues with early implementations of the NODF and wNODF methods. I strongly encourage the authors to reanalyze their data using the most current best-practice methods for measuring nestedness. Specifically, I recommend employing the normalized nestedness metric, NODF_c (Song et al., 2017). This new metric does not suffer from the statistical challenges associated with z-scores and, consequently, offers robustness in nestedness comparisons between networks (Song et al., 2017). Additionally, there is an R package (maxnodf) available (Hoeppke & Simmons, 2021) that facilitated the extraction of nestedness values with ease."

Experimental design

No comment.

Validity of the findings

No comment.

Additional comments

No comments.

---

## Round 0.3 · accepted · Accept

Dear Dr. Russo,

I am pleased to inform you that your manuscript has been accepted for publication in PeerJ! Congratulations!

Sincerely,
Daniel Silva

Reviewer 2 ·

Basic reporting

Thank you for your changes. I think they have greatly improved your manuscript. I did not see any problem with the rest of the paper and I think it is suitable for publication in its present form.

Experimental design

no comment

Validity of the findings

no comment

Additional comments

Congratulations for the much improved version.